

# Research on the correction method of hydraulic fracturing in-situ stress testing based on MLP-KFold

Yimin Liu[1], Huan Chen[3], Junchong Zhou[1] and Jinwu Luo[1*,2]

[1]School of Intelligent Manufacturing, Chengdu Technological University, Chengdu/611730, China
[2]School of Mechatronic Engineering, Southwest Petroleum University, Chengdu/610500, China
[3]Institute of Exploration Technology, CGS, Chengdu/611734, China

*Correspondence to*: Jinwu Luo (ljwcdtu@163.com) and Yimin Liu (153973418@qq.com)

**Abstract.** Hydraulic fracturing serves as a critical in-situ stress testing technique, where the accurate determination of rock fracture pressure and closure pressure in fracturing intervals is essential for precise in-situ stress estimation. During hydraulic

fracturing stress measurement, parameters including injection rate, viscosity, density, and compressibility ratio of fracturing fluid significantly affect the measurement accuracy of fracture and closure pressures, potentially introducing substantial errors in in-situ stress calculations. This study develops an MLP-KFold-based correction model for rock mechanical measurements by establishing a dataset derived from laboratory hydraulic fracturing simulations, incorporating fracturing fluid density, viscosity, injection rate, and corresponding rock fracture/closure pressures. Evaluation results demonstrate that the MLP-KFold

model achieves superior performance with a coefficient of determination ($R^2$=0.9937) on test sets, outperforming Random Forest ($\Delta$+1.89%), Support Vector Regression ($\Delta$+4.05%), and BiLSTM ($\Delta$+5.34%). Key error metrics including MAE (0.518), MSE (0.646), and maximum error (1.945MPa) remain at minimal levels. The model exhibits enhanced data utilization efficiency and evaluation stability with small-scale datasets while effectively preventing overfitting and improving generalization capabilities. Field applications demonstrate significant reduction in average percentage differences of calculated

stresses under different fracturing fluids ($\sigma$H: -21.48%, $\sigma$h: -29.03%), confirming its superior compensation effects. This research establishes a reliable compensation model for hydraulic fracturing pressures, providing an effective technical approach for correcting field measurement data and compensating in-situ stress calculation results, thereby contributing to the accurate assessment of regional stress profile states.

## 1 Introduction

The stress that is stored within the undisturbed rock mass is referred to as geo-stress or in-situ stress, which is caused by factors such as the self-weight of the rock and geological tectonic movements (Mcgarr and Gay, 1978; Amadei and Stephansson, 1997). Regional in-situ stress measurement and estimation have important applications in earthquake prediction research, underground engineering construction, mining, and oil and gas extraction. With the increasing demand for energy and mineral resources and the continuous intensification of mining efforts in China, shallow mineral resources are gradually diminishing,

and domestic mines are successively entering the stage of deep resource development. The "three highs" issues encountered



in deep mining (high in-situ stress, high temperature, and high water pressure) (Xie, 2019) will become the focus and difficulty in the study of deep mining rock mechanics (He et al., 2005). Therefore, accurately determining the in-situ stress state of the deep development area is a necessary approach to solving the above problems.

Observation and estimation of the in-situ stress state in the deep crust remain a major challenge in in-situ stress measurement.
Scientists have proposed dozens of in-situ stress testing methods, which can be classified into five categories based on data sources, as described in Table 1.

**Table.1: In-situ stress testing methods**

| No. | Method | Related literatures |
|---|---|---|
| 1 | Core-based method | Simmons et al., 1974; Siegfried and Simmons, 1978 |
| 2 | Borehole-based method | Scheidegger, 1962; Raleigh et al., 1976; Chandler, 1993; Fairhurst, 2003 |
| 3 | Geological method | Angelier, 1979; Hill et al., 1994; Adiyaman et al., 1998; Zoback, 2007 |
| 4 | Geophysical (or seismological) method | Crampin, 1985; Yale, 2003; Boness and Zoback, 2004 |
| 5 | Underground space-based method | Hill et al., 1994; Amadei and Stephansson, 1997 |

The hydraulic fracturing method, a subset of borehole-based techniques, is currently the only known approach capable of
directly measuring in-situ stress. Although theoretically unrestricted by depth, practical limitations—such as borehole conditions, testing technology, and temperature/pressure resistance of equipment—have resulted in very few successful hydraulic fracturing stress measurements worldwide at depths exceeding 1000m (Zhang and Stephansson, 2010; Chen et al., 2017). Consequently, precise measurement and estimation of in-situ stress using hydraulic fracturing remains a critical research challenge both domestically and internationally.
The hydraulic fracturing method features a relatively simple and rapid testing process, along with straightforward data processing and analysis. However, during testing, deformation of drill pipes and packers, as well as external factors related to the fluid mechanics parameters of fracturing fluids (e.g., viscosity, density, compressibility, and injection rate), can significantly affect the measurements of rock breakdown pressure, closure pressure, and reopening pressure in the fracturing interval. Consequently, these influences also introduce errors in subsequent calculations of the maximum and minimum
horizontal principal stresses, ultimately impairing the accurate estimation of in-situ stress. Related scholars have conducted in-depth research on the influence of fracturing fluid parameters such as flow rate, viscosity, and density on rock breakdown pressure and closure pressure. Ito (1991) and Chang (2014) proposed that the tensile strength of rock increases with the



injection rate. Zhou et al. (2013) and Zhang (2018) conducted laboratory hydraulic fracturing experiments, demonstrating that mud media with different densities significantly affect the measured values of rock breakdown. Matsunaga (1993) and Ishida

et al. (1997) confirmed in their studies on petroleum drilling that the viscosity of fracturing fluid influences rock breakdown. Wang (2012) and Zhou (2013) both used water as the fracturing fluid to analyze the effect of fluid compressibility on system compliance, which leads to errors in in-situ stress measurement. Ingrid and Marte (2017) employed a Bonded Particle Model (BPM) within the Discrete Element Method (DEM) framework to conduct coupled thermo-hydro-mechanical analysis, revealing that temperature gradients and fluid-rock compressibility ratios critically govern fracture dynamics in enhanced

geothermal systems: the coupled convective-conductive thermal effects shorten primary fractures and induce secondary microcracks, while cold fluid infiltration reduces near-wellbore pressure accumulation to delay propagation, with compressibility ratio governing fracture velocity and dynamic viscosity modulating thermal damage extent. Liu et al. (2019) optimized hydraulic fracturing simulation experiments using the uniform design method and preliminarily analyzed the influence of different fracturing fluid media on breakdown pressure through regression fitting. Guo et al. (2023) employed a

Generalized Regression Neural Network (GRNN) to predict breakdown pressure based on fluid scanning imaging logging data from the MH block of the Junggar Basin. The evaluation results showed that this model achieves high prediction accuracy, outperforming BP neural networks and the H-W model. Ma et al. (2024) utilized an LSTM to directly predict the breakdown pressure of horizontal wells in petroleum engineering, effectively establishing a nonlinear relationship between logging parameters and breakdown pressure in horizontal wells. Zou et al. (2024) investigated the influences of key parameters,

including rock temperature, in situ stress, injection rate, fluid viscosity, azimuth of the radial borehole, and the number of radial boreholes on the fracture morphology and breakdown pressure, the breakdown pressure of radial borehole fracturing can be reduced by 14.1%–43.7% compared to conventional fracturing. A higher fluid-rock temperature difference reduces the directional propagation range of fractures guided by the radial borehole. Increases in the vertical density of radial boreholes, injection rate, and fluid viscosity enhance the guiding ability of radial boreholes.

Measurement errors in rock breakdown pressure and closure pressure can lead to significant variations in the calculated maximum and minimum horizontal principal stresses, severely impacting the accuracy of hydraulic fracturing-based in-situ stress estimation (Wang et al., 2017). Accurately determining rock breakdown pressure and closure pressure has long been a challenging task. Reducing the errors in in-situ stress calculations caused by influencing factors such as fracturing fluid flow rate, viscosity, density, and compressibility—and establishing corresponding error compensation models—necessitates the

application of machine learning and deep learning methods. This paper constructed a dataset based on laboratory hydraulic fracturing simulation experiments. For relatively small-scale datasets, we employed a multi-layer perceptron (MLP) with K-fold cross-validation (KFold) to develop correction models for breakdown pressure and closure pressure. This approach demonstrates high data utilization efficiency and stable performance evaluation, exhibiting certain advantages over other machine learning models. The proposed method holds significant theoretical and practical value for precise regional stress

profile estimation, crustal stability assessment, earthquake prediction, and mine strata stability evaluation.





## 2 Principle of hydraulic fracturing in-situ stress testing

The classical hydraulic fracturing theory, based on the plane strain theory of elasticity, was proposed and refined by Haimson (1968). This theory is based on three key assumptions: First, rocks are considered to be homogeneous, linearly elastic, and isotropic materials, which means that the mechanical properties of rocks are the same in all directions, and there is a linear

relationship between stress and strain; second, rocks are assumed to be porous media, and the fluid flow within the pores follows Darcy's Law, which states that the fluid flow rate is directly proportional to the pressure gradient; finally, it is assumed that one of the principal axes of the in-situ stress is parallel to the borehole axis. Based on these assumptions, the fractures induced by hydraulic fracturing are vertical and perpendicular to the direction of the minimum horizontal principal stress, as shown in Figure 1.

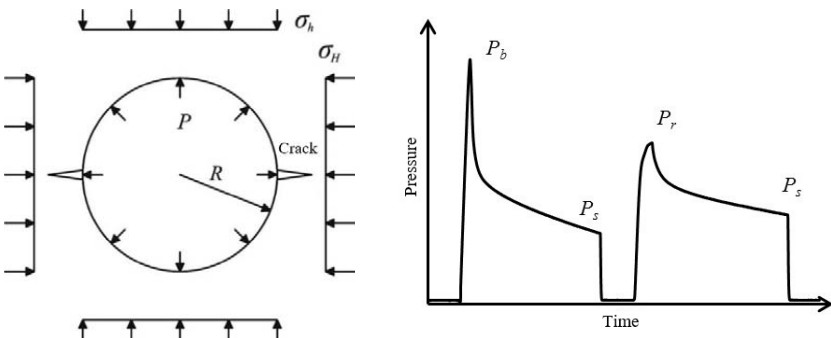


**Figure 1: Mechanical model and pressure curve diagram of hydraulic fracturing measurement**

Using the stress field model and fracture criteria depicted in Figure 1, the fracture values of the fractured rock section are shown below, according to the elastic theory (Timoshenko and Goodier, 1951):

$$P_b = 3\sigma_h - \sigma_H + T \tag{1}$$

$$P_s = \sigma_h \tag{2}$$

Here, $\sigma_H$ and $\sigma_h$ represent the maximum and minimum horizontal principal stresses, respectively, $P_b$ and $P_s$ represent the fracture pressure and closure pressure, respectively, and $T$ represents the tensile strength of the rock. Equations 1 and 2 indicate that the fracture values of the rock are independent of the size of the borehole and the rock's elastic modulus, and are mainly determined by the tensile strength of the rock and the magnitude of the in-situ stress around the borehole. Therefore, accurately

obtaining the fracture pressure, closure pressure, and tensile strength of the rock is key to improving the accuracy of hydraulic fracturing stress measurements.



## 3 Construction of hydraulic fracturing dataset

### 3.1 Design of simulation experiment

To simulate the mechanical process of hydraulic fracturing testing, this study employs the hollow-cylinder test method (Wang, 2014). Granite core samples with a diameter of 94 mm were prepared into test specimens (diameter-to-height ratio 1:1.6, height 150 mm), featuring a 20 mm central borehole. To minimize size effects, the specimen dimensions exceed 10 times the maximum mineral grain size (Cai, 2002). For the application of planar confining pressure, the cylindrical rock core was encased in cement mortar to form a cuboidal specimen (Fig. 2). The rock samples exhibit good integrity, with quartz and amphibole content ranging between 20% and 40%, and no pre-existing joints. The parameters are summarized in Table 2.

**Table.2: Parameters table of experimental cores**

| Lithology | Diameter (mm) | Height (mm) | Ratio of diameter to height | Center pore-diameter (mm) |
|---|---|---|---|---|
| Granite | 94 | 150 | 1:1.6 | 20 |

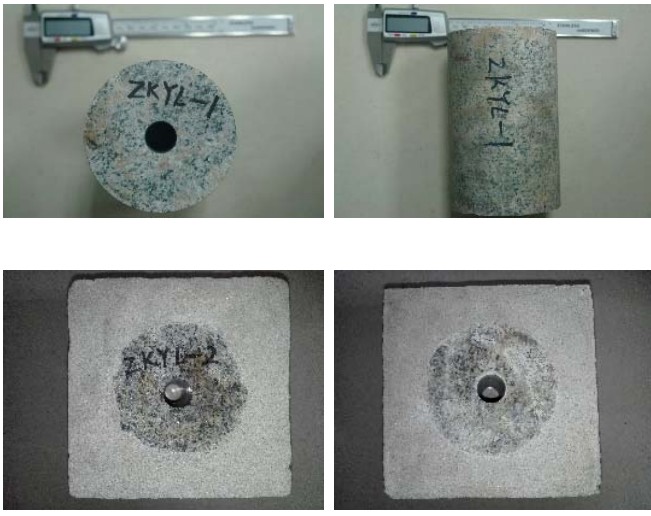

**Figure 2: Photo of hydraulic fracturing cuboidal specimen**



The fracturing fluids commonly used in hydraulic fracturing mainly include water-based, oil-based, foam-based and slurry-based types (Cuisiat and Haimson, 1992; Birdsell et al., 2015), with their key parameters listed in Table 3.

**Table.3: In-situ stress testing methods**

| No. | Category | Parameters |
|---|---|---|
| 1 | Aqueous fracturing fluids | Water (density 1 g/cm³, viscosity 1 mPa·s) or sodium carboxymethyl cellulose (CMC) solution (adjustable viscosity) |
| 2 | Oil-based fracturing fluids | Crude oil or diesel as base fluid, supplemented with thickeners (e.g., mineral oil, density 0.88 g/cm³, viscosity 130.6 mPa·s) |
| 3 | Foam fracturing fluids | Composed of gas ($N_2$/$CO_2$), liquid phase (aqueous/oil-based), and foaming agents |
| 4 | Slurry medium | Contains clay minerals, weighting materials, and chemical additives, with adjustable density and viscosity |

The simulation experiments employed water-based, oil-based and mud as the fracturing fluids media in this paper, with the
injection rate controlled by adjusting the pump flow rate. The experimental factors and their corresponding levels are summarized in Table 4.

**Table 4: Experimental factors and their levels**

| Factor | Level | Unit | Parameter value |
|---|---|---|---|
| Injection rate | 8 | MPa/s | 0.17; 0.35; 0.48; 0.55; 0.6; 0.69; 3.15; 3.88; 4.25 |
| Density | 3 | g/cm³ | 0.88; 1.01; 1.51 |
| Viscosity | 5 | mPa•s | 1; 70.2; 130.6; 171.6; 284.5 |

**3.2 Dataset construction**

For 35 cuboidal specimens with different tensile strengths, hydraulic fracturing simulation experiments were conducted 35
times based on an 8-level injection rate, a 2-level density, and a 5-level viscosity (as shown in Table 2). The experimental results (fracturing pressure and closure pressure) were obtained and presented in Table 5.



**Table 5: Optimal design table of values for the hydraulic fracturing simulation experiments**

| No. | Influencing factors | | | | Experimental results | |
| --- | --- | --- | --- | --- | --- | --- |
| | Density | Viscosity | Injection Rate | Tensile strength | Fracturing pressure | Closure pressure |
| 1 | 1.01 | 1 | 0.69 | 14.42 | 18.7 | 2.14 |
| 2 | 1.01 | 1 | 1.2 | 12.92 | 18.0 | 2.54 |
| 3 | 1.01 | 1 | 1.2 | 9.48 | 22.0 | 6.26 |
| 4 | 1.01 | 1 | 0.69 | 8.3 | 29.0 | 10.35 |
| 5 | 1.01 | 1 | 0.69 | 7.6 | 29.0 | 10.7 |
| 6 | 1.01 | 1 | 0.69 | 6.38 | 28.0 | 10.81 |
| 7 | 1.01 | 1 | 0.55 | 4.5 | 32.1 | 13.8 |
| 8 | 1.01 | 1 | 0.69 | 2.8 | 31.0 | 14.1 |
| 9 | 1.01 | 1 | 0.69 | 3.06 | 31.5 | 14.22 |
| 10 | 1.01 | 1 | 4.25 | 16.3 | 24.1 | 3.9 |
| 11 | 1.01 | 1 | 3.88 | 18.4 | 27.0 | 4.3 |
| 12 | 1.01 | 1 | 3.15 | 18.8 | 30.0 | 5.6 |
| 13 | 1.01 | 1 | 4.25 | 16.6 | 28.2 | 5.8 |
| 14 | 1.01 | 1 | 3.88 | 6.6 | 25.0 | 9.2 |
| 15 | 1.01 | 1 | 3.88 | 15 | 36.0 | 10.5 |
| 16 | 1.01 | 1 | 3.88 | 11.82 | 34.0 | 11.09 |
| 17 | 1.01 | 1 | 3.15 | 18.2 | 48.0 | 14.9 |
| 18 | 1.01 | 1 | 4.25 | 8 | 38.0 | 15 |
| 19 | 1.01 | 1 | 3.88 | 13.2 | 44.0 | 15.4 |
| 20 | 1.01 | 1 | 0.48 | 9.4 | 8.22 | 0 |



| 21 | 1.01 | 1 | 0.48 | 9.76 | 10.95 | 1.19 |
|----|------|------|------|-------|-------|------|
| 22 | 1.01 | 1 | 0.48 | 7.65 | 8.84 | 1.19 |
| 23 | 1.01 | 70.2 | 0.6 | 8.99 | 10.17 | 1.18 |
| 24 | 1.01 | 70.2 | 0.6 | 5.85 | 7.03 | 1.18 |
| 25 | 1.01 | 70.2 | 0.6 | 8.33 | 9.5 | 1.18 |
| 26 | 0.88 | 130.6 | 0.55 | 11.88 | 13.07 | 1.19 |
| 27 | 0.88 | 130.6 | 0.55 | 12.6 | 13.78 | 1.18 |
| 28 | 0.88 | 130.6 | 0.55 | 9.98 | 11.22 | 1.19 |
| 29 | 1.51 | 171.6 | 0.17 | 9.57 | 12.74 | 1.87 |
| 30 | 1.51 | 171.6 | 0.17 | 9.07 | 12.25 | 1.88 |
| 31 | 1.51 | 171.6 | 0.17 | 10.37 | 13.54 | 1.87 |
| 32 | 1.51 | 284.5 | 0.35 | 9.64 | 12.83 | 1.89 |
| 33 | 1.51 | 284.5 | 0.35 | 12.31 | 15.47 | 1.89 |
| 34 | 1.51 | 284.5 | 0.35 | 9.83 | 12.99 | 1.89 |
| 35 | 1.51 | 284.5 | 0.35 | 12.27 | 15.44 | 1.89 |

### 3.3 Correlation analysis

Dual analytical methodologies—the Pearson correlation coefficient and SHAP (SHapley Additive exPlanations) value
heatmaps—were systematically employed to quantify variable interdependencies. The Pearson metric provides efficient
identification of linear correlations, enabling preliminary feature screening, while SHAP decomposition elucidates complex
feature contributions within the model architecture, particularly nonlinear interactions (Nahler, 2020). Figure 3 quantitatively
illustrates the operational relationships between extrinsic parameters (injection rate, density, and viscosity) and critical
geomechanical outputs: fracturing pressure ($P_b$) and closure pressure ($P_s$).



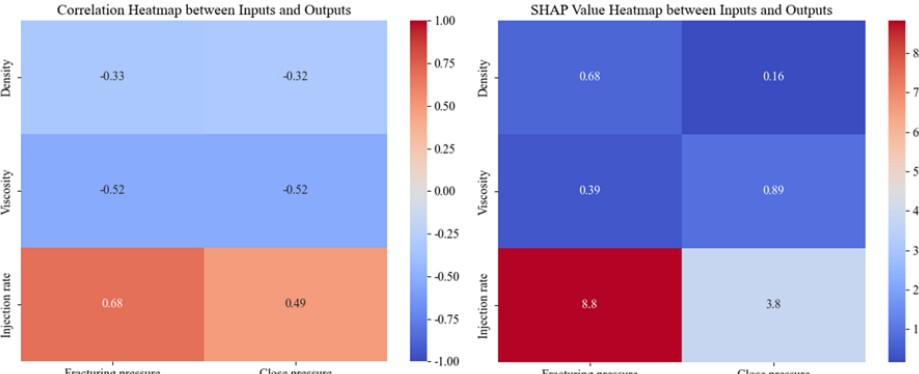


**Figure3. Correlation between Inputs and Outputs**

Figure 3 demonstrates that injection rate exhibits the most significant influence on fracturing pressure ($P_b$) and closure pressure ($P_s$), with a pronounced positive correlation observed, particularly for fracture pressure. In contrast, fluid density and viscosity demonstrate comparatively weaker correlations with these output parameters. Owing to the multifaceted influences on $P_b$ and
$P_s$—where complex interactions among governing factors may involve nonlinear relationships—conventional linear regression models may prove inadequate to accurately characterize these dependencies. To enhance predictive accuracy, this study proposes the adoption of neural network architectures or machine learning frameworks to develop error-compensated predictive models. Such approaches are anticipated to better capture the inherent nonlinear dynamics between operational variables and geomechanical responses, thereby optimizing pressure prediction fidelity in hydraulic fracturing operations.

**4 MLP-KFold model**

**4.1 MLP model**

The multilayer perceptron (MLP) model is a fully connected neural network composed of multiple neurons. By adjusting the weights of these neurons, the model minimizes prediction errors, enabling effective training and subsequent outcome prediction (Zhang et al., 2021). The MLP features a multi-layered structure, including an input layer, one or more hidden layers,
and an output layer, as illustrated in the network architecture diagram (Figure 4).



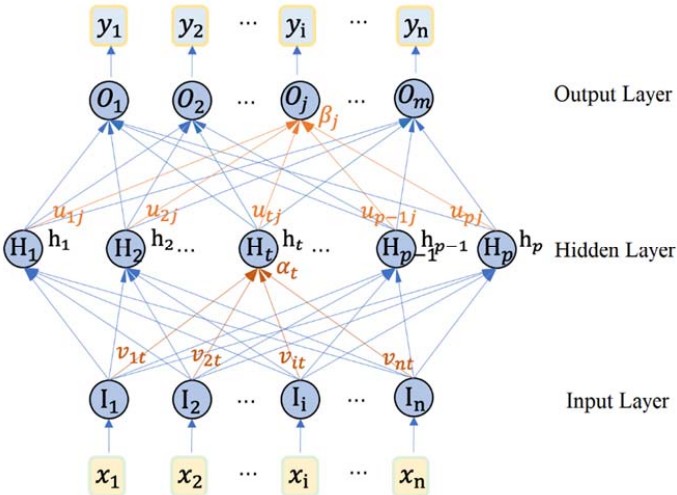

**Figure 4: MLP Model Structure Diagram**

The neurons in an MLP model receive input signals, sum them with weights, and produce an output through an activation function. Building upon the perceptron model, MLP increases the nesting level of neurons and introduces an activation function
between the inputs and outputs of each layer, thereby enhancing the learning capabilities of the MLP model. The output formula of a neuron is shown in Equation 3, where $\sigma$ represents the activation function.

$$y = \sigma\left(\sum_{i=1}^{n}\omega_i x_i + b\right) \tag{3}$$

And from the hidden layer to the output layer: suppose there are $p$ neurons in the output layer, and the weight matrix from the hidden layer to the output layer is $W_2$. Then the output of the output layer is shown in Equation 4, where $k = 1, 2, \ldots, p$. By
continuously iterating and updating the weights and biases, the model can effectively fit the data and make accurate predictions.

$$\hat{y}_k = \sigma\left(\sum_{j=1}^{n}W_{2kj}h_j + b_{2k}\right) \tag{4}$$





## 4.2 Model Structure of MLP-KFold

The MLP-KFold model combines a Multilayer Perceptron (MLP) with K-Fold Cross-validation (KFold) as a method for model training and evaluation. The MLP-KFold is built using the Sequential model, which is composed of multiple network layers stacked in sequence. The data flows through each layer in order from front to back for processing. As shown in Figure 5, the MLP model is primarily a feedforward neural network composed of an input layer, hidden layers, a Dropout layer, and an output layer. The input layer has a dimension of 6, corresponding to the number of features in the dataset (as listed in Table 2). The three hidden layers consist of 96, 48, and 24 neurons, respectively, all using the ReLU activation function and L2 regularization to prevent overfitting. A Dropout layer is added after the first and second hidden layers to further mitigate the risk of overfitting. The output layer contains 2 neurons, corresponding to the two prediction targets: fracturing pressure ($P_s$) and closure pressure ($P_b$).

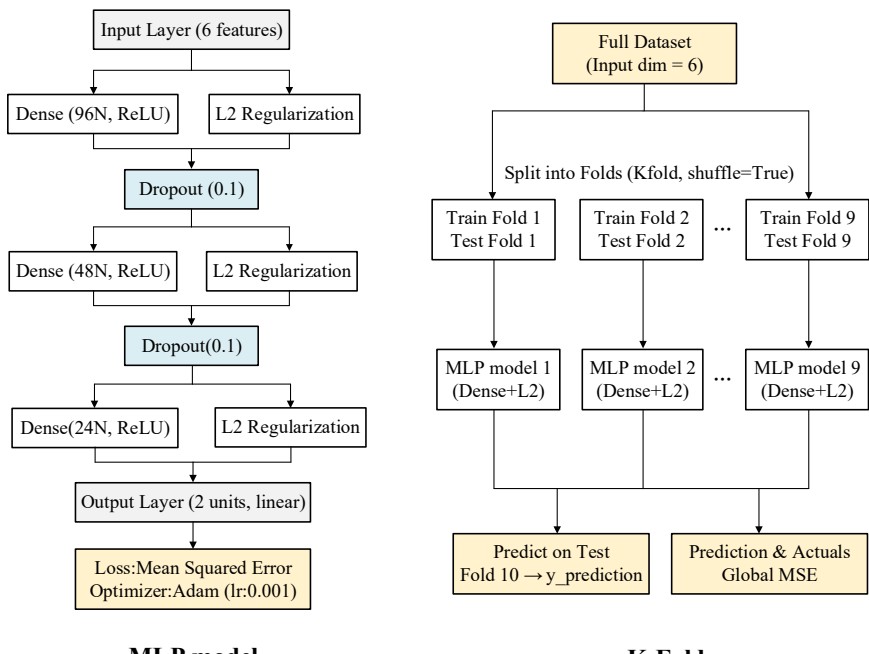

**Figure 5: MLP-KFold Model Structure Diagram**



To maximize the utility of the small-scale dataset and enhance evaluation reliability, the K-Fold model adopts 10-fold cross-
validation (KFold), therefore, it is suitable for the smaller scale dataset in this paper. The data is randomly split into 10
subsets—9 for training and 1 for testing—and this process repeats 10 times, cycling through each subset as the test set. This
ensures every data point contributes to validation, leading to a more robust and stable performance assessment.

### 4.3 Parameter Setting

The core and training parameters of the MLP-KFold model are systematically configured prior to model implementation. The
core parameters include: input dimension, output units, maximum number of neurons, and dropout. The training parameters
include: number of epochs, learning rate, batch size, and validation ratio, as detailed in Table 6. Following parameter
initialization, the model was trained using the prepared training dataset through iterative optimization processes.

**Table 6: Parameter Setting Table**

|  | Parameter | Value |
| --- | --- | --- |
| Core parameters | input dimension | 6 |
|  | output units | 2 |
|  | maximum number of neurons | 96 |
|  | dropout | 0.05 |
| Training parameters | epochs | 100~150 |
|  | learning rate | Adam optimizer, 0.001 |
|  | batch size | 1 |
|  | validation ratio | 0.1 |

### 5 Discussion

### 5.1 The prediction performance of MLP KFold

Following the model architecture and parameter configuration detailed in Chapter 3, the machine learning model was trained
using the specified training dataset (refer to Table 5). The training process enables the model to learn inherent patterns and
characteristic features within the data, thereby establishing predictive capabilities for subsequent correction and forecasting
applications. Post-training visualization analysis revealed the comparative performance between actual and predicted values
for both fracture pressure and closure pressure, as illustrated in Figure 6. In this graphical representation, blue bars denote
measured pressure values while red bars indicate model-predicted values, demonstrating the algorithm's predictive accuracy
through visual comparison of these dual pressure parameters.



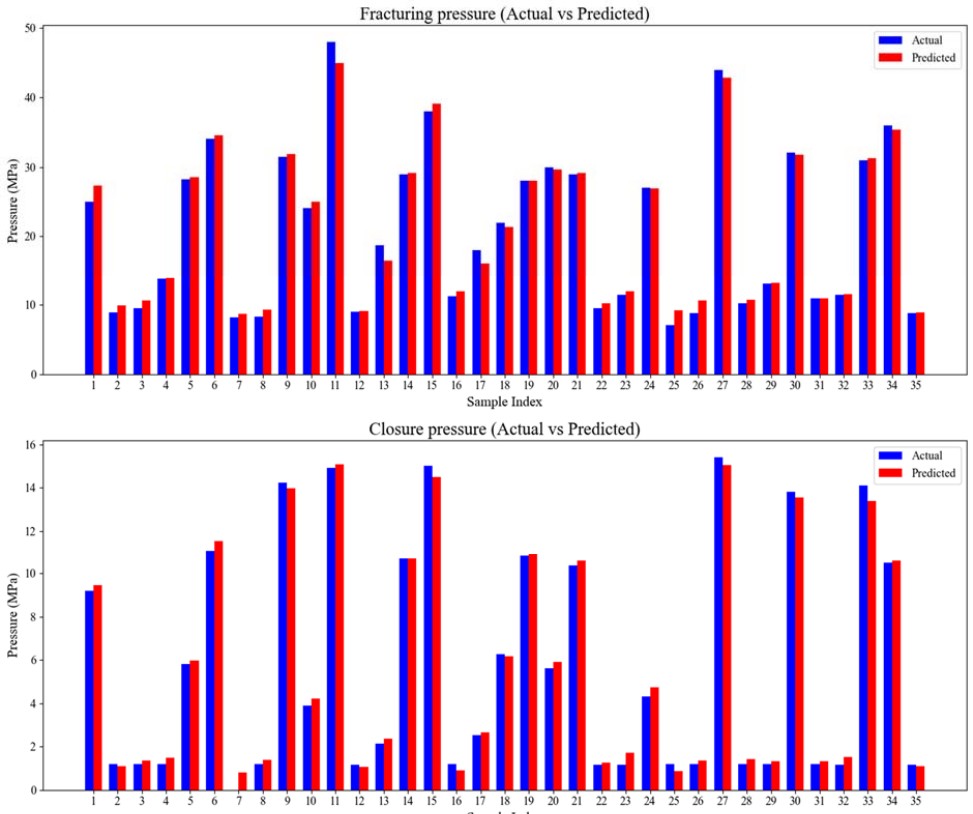

**Figure 6: Comparative performance between actual and predicted values for both fracture pressure and closure**

**pressure**

As evidenced in Table 7, the MLP-KFold model demonstrates exceptional predictive accuracy, achieving a mean $R^2$ coefficient of determination of 0.9937 – a value remarkably close to the ideal unity. The error metrics further substantiate this performance, with the Mean Absolute Percentage Error (MAPE) and Mean Squared Error (MSE) registering at minimal values of 4.115% and 0.6457 respectively. Notably, the maximum observed prediction error remains constrained to 1.9449 MPa, confirming

tight error distribution boundaries. These collective findings indicate that the proposed model successfully accounts for 99.37% of total data variance, achieving near-perfect goodness-of-fit. The minimal divergence between predicted values and empirical observations validates the model's robust generalization capabilities across the experimental dataset.



**Table 7: Performance indices of the MLP-KFold model**

| Outputs | R² | MSE | RMSE | MAE | MAPE | Max Error |
|---|---|---|---|---|---|---|
| Fracturing pressure | 0.9910 | 1.1918 | 1.0917 | 0.7758 | 5.19% | 3.0725 |
| Closure pressure | 0.9964 | 0.0995 | 0.3155 | 0.2597 | 3.04% | 0.8173 |

**5.2 Multi-model comparative analysis**

Based on the satisfactory fitting performance achieved by the MLP-KFold model, a systematic comparison study is subsequently conducted to evaluate its predictive efficacy against alternative machine learning architectures, including Random Forest (ensemble-based decision tree model), Support Vector Regression (SVR, kernel method), and Bidirectional Long Short-Term Memory (BiLSTM, deep sequential learning framework). This benchmarking framework employs identical training datasets and preprocessing protocols to ensure fair performance assessment. Quantitative metrics such as mean squared

error (MSE) and coefficient of determination (R²) will be comparatively analyzed across all models, while their generalization capabilities and computational efficiency will be critically examined. The cross-model comparison aims to (1) validate the robustness of MLP-KFold in handling geomechanical pressure prediction tasks, (2) identify algorithm-specific advantages under controlled experimental conditions, and (3) establish methodological guidelines for optimal model selection in fracture pressure characterization studies. Table 8 shows performance indices of these multi-model.

**Table 8: Performance indices table of the the multi-model**

| Model | Outputs | R² | MSE | RMSE | MAE | MAPE | Max Error |
|---|---|---|---|---|---|---|---|
| MLP-KFold | Fracturing pressure | 0.9910 | 1.0695 | 0.7976 | 0.5378 | 5.19% | 3.0725 |
| | Closure pressure | 0.9964 | 0.0995 | 0.3155 | 0.2597 | 3.04% | 0.8173 |
| Random Forest | Fracturing pressure | 0.9643 | 4.7345 | 2.1759 | 1.3670 | 6.48% | 8.3760 |
| | Closure pressure | 0.9855 | 0.4058 | 0.6370 | 0.3495 | 4.62% | 2.6518 |
| SVR | Fracturing pressure | 0.9427 | 7.6026 | 2.7573 | 1.5770 | 8.54% | 13.1444 |
| | Closure pressure | 0.9642 | 1.0007 | 1.0003 | 0.7034 | 6.85% | 3.4494 |
| BiLSTM | Fracturing pressure | 0.9658 | 3.0339 | 1.7418 | 1.4123 | 14.78% | 3.2565 |
| | Closure pressure | 0.9154 | 0.3963 | 0.6295 | 0.4900 | 6.18% | 1.3806 |

The radar chart of error indices is plotted using Z-score normalization to provide a more intuitive visualization of the performance of different models across various error indices, facilitating model comparison and evaluation. Since five error coefficients from Table 8 needed to be compared simultaneously, Z-score standardization is applied to effectively mitigate the influence of scale differences and outliers in the radar chart.





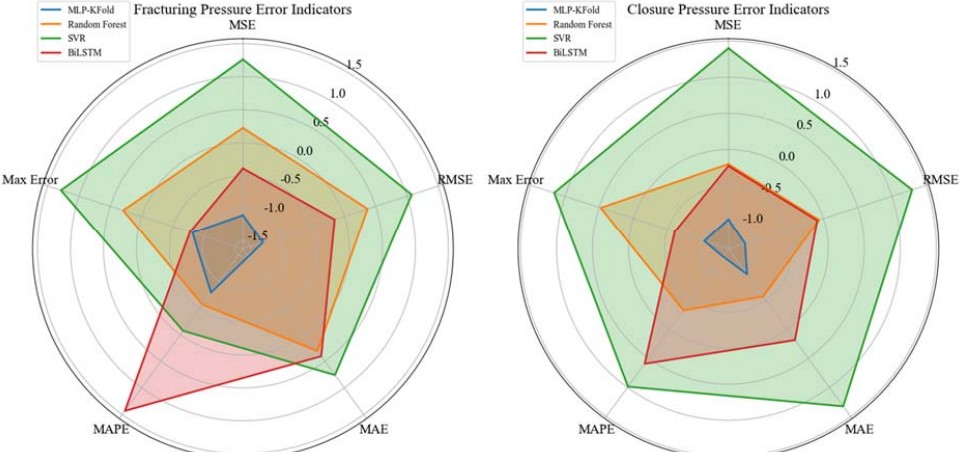


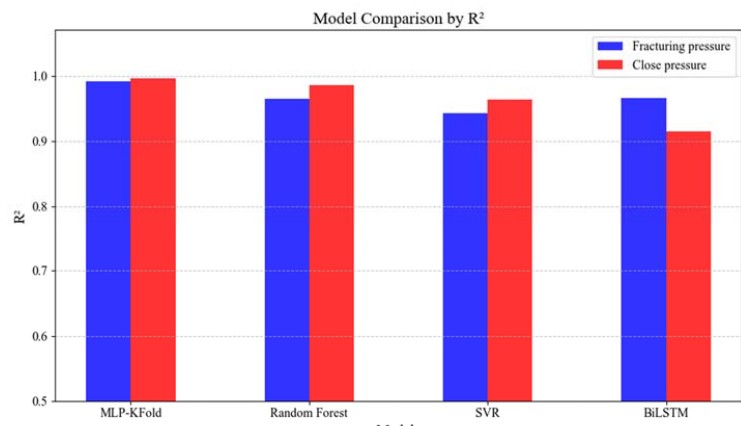

**Fig 7. Comparison chart of evaluation indices of the multi-model**

In the context of small-scale datasets, Figure 7 shows that selecting an appropriate model is critical, as limited data inherently increases the risk of model overfitting and reduces generalization capability. MLP-KFold demonstrates robust performance

when applied to small-scale hydraulic fracturing simulation datasets, and its effectiveness can be attributed to the following aspects:



(1) Cross-validation mechanism: The K-fold cross-validation integrated in MLP-KFold enables full utilization of limited data, thereby enhancing model generalization and mitigating overfitting risks. By iteratively partitioning the dataset into training and validation subsets, this approach ensures reliable performance evaluation while maximizing data exploitation.

(2) Simplified model architecture: As a relatively simple neural network structure, the Multilayer Perceptron (MLP) inherently requires fewer data samples to achieve stable convergence compared to complex deep learning architectures. This characteristic makes MLP particularly suitable for small-scale datasets where intricate pattern learning is constrained by data scarcity.

(3) Regularization integration: MLP-KFold systematically incorporates regularization techniques, including L2 regularization

and Dropout, to further suppress overfitting tendencies. These mechanisms impose constraints on weight optimization and randomly deactivate neurons during training, effectively reducing model complexity and enhancing robustness to noise in limited data scenarios.

This combination of methodological advantages positions MLP-KFold as a computationally efficient and statistically reliable framework for analyzing small-scale experimental datasets in hydraulic fracturing simulations.

**6 Engineering application**

The MSZK and ZPZK boreholes are situated in Mengshan County, Guangxi, China, serving as adjacent boreholes for geostress testing within a hydraulic investigation and design project. Both boreholes were designed to a depth of 275 m, with hydraulic fracturing in-situ stress measurements conducted following standardized operational procedures. A total of 11 fracturing intervals were implemented within the depth range of 145.2–244.3 m across both boreholes, with key fracturing parameters

and results summarized in Table 8. The MSZK maintained favorable wellbore conditions, enabling the use of clean water as fracturing fluid. In contrast, the ZPZK required drilling mud (density: 1.5 g/cm³, viscosity: 235 mPa·s) for wall stabilization due to severe borehole collapse. A comparative analysis of hydraulic fracturing stress measurement outcomes between MSZK and ZPZK under distinct fracturing fluid conditions is presented in Table 8, demonstrating significant differences in in-situ stress measurement outcomes attributable to fluid medium variations.

**Table 8: Comparison table of hydraulic fracturing in-situ stress measurement results**

| Depth(m) | Fracturing parameters (MPa) | | | | | | Stress Value (MPa) | | | | |
|---|---|---|---|---|---|---|---|---|---|---|---|
| | $P_b$(m) | $P_s$(m) | $T_m$ | $P_b$(w) | $P_s$(w) | $T_w$ | $\sigma_H$(m) | $\sigma_h$(m) | $\sigma_H$(w) | $\sigma_h$(w) | $\sigma_v$ |
| 145.2~146.7 | 22.33 | 8.62 | 10.59 | 13.21 | 7.34 | 7.59 | 12.93 | 8.62 | 12.8 | 7.34 | 37.92 |
| 149.8~151.3 | 16.77 | 8.75 | 4.26 | 14.1 | 7.86 | 4.26 | 12.53 | 8.75 | 12.57 | 7.86 | 39.08 |
| 157.0~158.5 | 21.55 | 12.6 | 6.21 | 13.17 | 8.82 | 6.21 | 21.24 | 12.6 | 14.6 | 8.82 | 40.95 |
| 163.4~164.9 | 14.67 | 10.6 | 7.78 | 11.84 | 7.12 | 7.78 | 18.58 | 10.6 | 12.29 | 7.12 | 42.62 |
| 172.3~173.8 | 30.4 | 12.3 | 14.23 | 12.03 | 6.13 | 14.23 | 19.24 | 12.3 | 10.62 | 6.13 | 44.93 |



| 182.0~183.5 | 29.49 | 12.67 | 16.25 | 9.71 | 5.84 | 16.25 | 23.22 | 12.67 | 10.37 | 5.84 | 47.45 |
| 200.6~202.1 | 29.45 | 12.92 | 15.49 | 12.95 | 8.54 | 15.49 | 23.18 | 12.92 | 14.73 | 8.54 | 52.38 |
| 214.6~216.1 | 30.76 | 14.81 | 16.19 | 9.3 | 7.20 | 16.19 | 28.15 | 14.81 | 12.33 | 7.20 | 55.91 |
| 220.5~222.0 | 30.27 | 18.37 | 10.8 | 11.08 | 7.91 | 10.8 | 33.89 | 18.37 | 13.25 | 7.91 | 57.45 |
| 235.1~236.6 | 29.51 | 20.68 | 10.92 | 14.08 | 9.37 | 10.92 | 31.58 | 20.68 | 16.88 | 9.37 | 61.24 |
| 242.8~244.3 | 28.63 | 16.92 | 11.89 | 12.12 | 8.68 | 11.89 | 32.06 | 16.92 | 15.06 | 8.68 | 63.24 |

Here, $P_b$(m): ZPZK fracturing pressure by mud; $P_s$(m): ZPZK closure pressure by mud; T: rock tensile strength; $P_b$(w): MSZK fracturing pressure by water; $P_s$(w): MSZK closure pressure by water; $\sigma_H$(m): ZPZK maximum horizontal principal stress; $\sigma_h$(m): ZPZK minimum horizontal principal stress; $\sigma_H$(w): MSZK maximum horizontal principal stress; $\sigma_h$(w): MSZK minimum horizontal principal stress; $\sigma_v$: vertical principal stress (the overburden rock unit weight was assigned as 26.5 kN/m³).

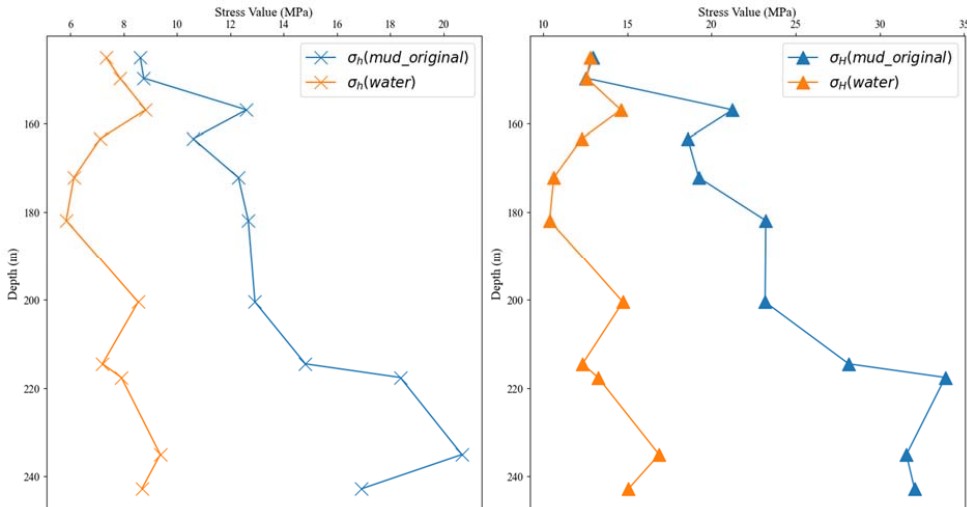

**Figure 8: Comparative curves of maximum and minimum horizontal principal stresses between ZKZP(mud) and MSZK(water)**

Figure 8 demonstrates the comparative curves of maximum and minimum horizontal principal stresses between ZKZP(mud) and MSZK(water). The analysis reveals significantly higher in-situ stress calculation results for ZKZP compared to MSZK. The average percentage differences reach 39.32% for maximum horizontal principal stress and 39.61% for minimum horizontal principal stress within equivalent depth intervals. Considering the close proximity of these two boreholes (only 50m



apart) and their comparable geological conditions with identical lithological characteristics in surrounding strata, this substantial discrepancy strongly suggests that the drilling mud medium in boreholes exerts considerable influence on

measurement outcomes such as rock breakdown values. To address this systematic bias, the MLP-KFold model was subsequently employed to calibrate critical output parameters (fracture pressure and closure pressure values). The refined in-situ stress calculation results after correction are presented in Table 9.

**Table 9: Comparison table of hydraulic fracturing in-situ stress measurement results after compensation**

| Depth(m) | Fracturing Parameters(MPa) | | | | | | Stress Value (MPa) | | | | |
|---|---|---|---|---|---|---|---|---|---|---|---|
| | $P_b$(m_c) | $P_s$(m_c) | $T_m$ | $P_b$(w) | $P_s$(w) | $T_w$ | $\sigma_H$(m_c) | $\sigma_h$(m_c) | $\sigma_H$(w) | $\sigma_h$(w) | $\sigma_v$ |
| 145.2~146.7 | 14.1 | 6.81 | 10.59 | 13.21 | 7.34 | 7.59 | 13.92 | 6.81 | 12.8 | 7.34 | 37.92 |
| 149.8~151.3 | 12.23 | 7.52 | 4.26 | 14.1 | 7.86 | 4.26 | 14.59 | 7.52 | 12.57 | 7.86 | 39.08 |
| 157.0~158.5 | 17.18 | 7.86 | 6.21 | 13.17 | 8.82 | 6.21 | 12.61 | 7.86 | 14.6 | 8.82 | 40.95 |
| 163.4~164.9 | 15.38 | 5.79 | 7.78 | 11.84 | 7.12 | 7.78 | 9.77 | 5.79 | 12.29 | 7.12 | 42.62 |
| 172.3~173.8 | 18.82 | 7.47 | 14.23 | 12.03 | 6.13 | 14.23 | 17.82 | 7.47 | 10.62 | 6.13 | 44.93 |
| 182.0~183.5 | 18.63 | 5.76 | 16.25 | 9.71 | 5.84 | 16.25 | 14.9 | 5.76 | 10.37 | 5.84 | 47.45 |
| 200.6~202.1 | 20.81 | 6.97 | 15.49 | 12.95 | 8.54 | 15.49 | 15.59 | 6.97 | 14.73 | 8.54 | 52.38 |
| 214.6~216.1 | 20.44 | 6.73 | 16.19 | 9.3 | 7.20 | 16.19 | 15.94 | 6.73 | 12.33 | 7.20 | 55.91 |
| 220.5~222.0 | 20.87 | 8.55 | 10.8 | 11.08 | 7.91 | 10.8 | 15.58 | 8.55 | 13.25 | 7.91 | 57.45 |
| 235.1~236.6 | 22.97 | 9.29 | 10.92 | 14.08 | 9.37 | 10.92 | 15.82 | 9.29 | 16.88 | 9.37 | 61.24 |
| 242.8~244.3 | 20.7 | 7.82 | 11.89 | 12.12 | 8.68 | 11.89 | 14.65 | 7.82 | 15.06 | 8.68 | 63.24 |

Here, (m_c) represents the pressure value of mud as the fracturing fluid after the MLP-KFold correction. The comparative curves of maximum and minimum horizontal principal stresses between the calibrated ZKZP and MSZK are presented in Figure 9. As illustrated in Figure 9, the discrepancy in calculated in-situ stress values between the two boreholes shows a marked reduction after model calibration. Within equivalent depth intervals, the maximum horizontal principal stress difference decreases to 17.84%, while the minimum horizontal principal stress difference demonstrates a more pronounced

improvement, achieving a remarkable decrease to 10.58%.




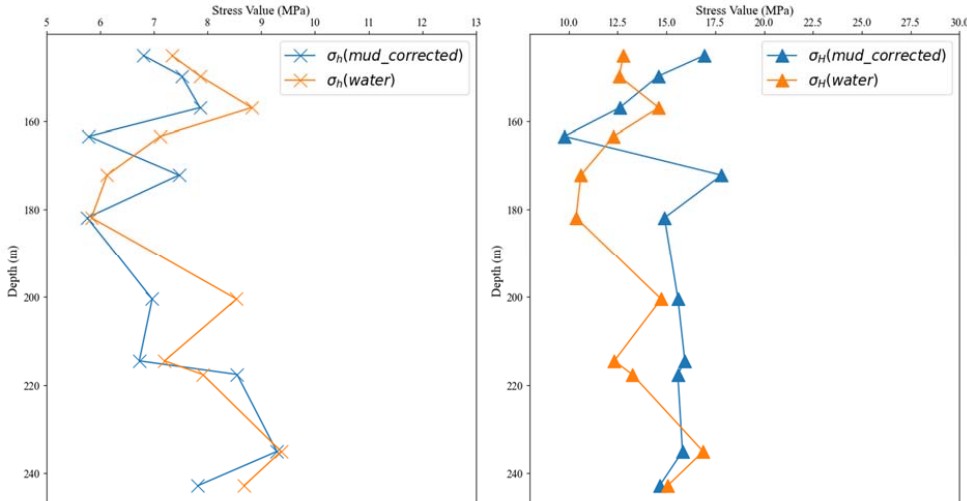

**Figure 9: The comparative curves of maximum and minimum horizontal principal stresses between the calibrated ZKZP and MSZK**

### 7 Conclusions

Based on the dataset constructed from simulation experiments of hydraulic fracturing, this paper establishes a rock mechanics measurement result correction model based on MLP KFold for rock mechanical measurements. Model evaluation and engineering applications demonstrate its exceptional performance in compensating for measurement deviations of hydraulic fracturing-induced fracture and closure pressures, thereby improving the accuracy of in-situ stress calculations. The integration of MLP and K-Fold cross-validation establishes a data-efficient framework for in-situ stress measurement correction,

providing a standardized technical solution for refining field measurement data and optimizing regional stress profile assessments. The main conclusions are as follows:

(1) The MLP-KFold model achieves a coefficient of determination ($R^2$=0.9937) on test datasets, outperforming benchmark models (Random Forest, Support Vector Regression, BiLSTM) by $\Delta$+1.89%–5.34%, highlighting its predictive superiority. Its minimal error metrics—mean absolute error (MAE=0.518), mean squared error (MSE=0.646), and maximum error (1.945

MPa)—validate the model's robustness and generalization capability with limited experimental data.

(2) Field applications reveal that the model significantly reduces average discrepancies in principal stress calculations under varying fracturing fluid conditions ($\sigma_H$: -21.48%; $\sigma_h$: -29.03%), effectively mitigating errors caused by differences in fracturing fluid properties (density, viscosity, or injection rate). This conclusively demonstrates the model's engineering applicability



and effectiveness in hydraulic fracturing stress measurements under complex field conditions, including geological
heterogeneity and time-varying operational parameters.

**Data Availability**

The data used to support the findings of this study are available from the corresponding author upon request.

**Author contribution**

Yimin Liu: writing the initial draft of the manuscript, conducting data analysis, and establishing the correction model; Jinwu
Luo: manuscript validation, experimental data acquisition and analysis; Huan Chen: simulation experimental data acquisition
and model establishment; Junchong Zhou: simulation experiment design and data acquisition.

**Conflicts of Interest**

The authors declare that there are no conflicts of interest regarding the publication of this paper, and the authors confirm that
the mentioned received funding in the "Acknowledgment" section did not lead to any conflict of interests regarding the
publication of this manuscript.

**Acknowledgments**

This work was supported by Project funded by China Postdoctoral Science Foundation (2019M650782), National Natural
Science Youth Foundation of China (41804089), and Project of Observation Instrument Development for Integrated
Geophysical Field of China Mainland (Y201802).

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
