# Peer review of "Research on the correction method of hydraulic fracturing in-situ stress testing based on MLP-KFold"

_EGUsphere, 2025_

## Author Comment (AC2)

Dear reviewer 1#:

We are very grateful to your comments, and we have carefully read and considered the referee's comments, and these comments are important for improving the quality of this manuscript. Based on these comments, we have made carefully modification and proofreading on the original manuscript, the revised parts have been marked in red in revised version, and the detail modifications are shown in next chapter.

Thank you very much for your suggestion and consideration, and we look forward to hearing from you.

Best regards,

Yimin Liu and Jinwu Luo.

Detailed revision:

1. Some references are outdated, slightly lacking an understanding of the latest research progress. I suggest adding some latest literature related to this study.

Modification: In the revised manuscript, a new reference to Liu et al. (2024) has been added to the literature review section. This latest reference is particularly relevant to our research as it introduces an advanced in-situ stress calculation model, and addition strengthens our literature review by providing a more precise and up-to-date method for in-situ stress calculation. And Ma et al. (2024) and Zou et al. (2024) in the first draft also reflect the real-time and relevance of literature review.

Liu J, Cheng Y, Shu H, et al. 2024. Geostress Calculation Model of Horizontal Hole Hydraulic Fracturing Method Considering Different Fracturing Fluid Flow Rates [J]. Yellow River, 46(12):131-136. (abstract in English)

2. Insufficient Dataset Scale and Diversity.

Explanation and modification: Thank you for your valuable feedback and suggestions regarding the scale and diversity of our dataset. We fully acknowledge the importance of a diverse and sufficiently large dataset in ensuring the comprehensive applicability of our model. In our study, we have utilized 35 sets of hydraulic fracturing experimental data. It's crucial to note that hydraulic fracturing is a destructive testing method, and the success rate of such experiments is relatively low. Each successful experiment requires significant time and resources, including the preparation of rock cores and the careful calibration of experimental conditions. Given these constraints, obtaining 35 valid datasets represents a substantial effort and a rare accumulation of data in this field.

The primary focus of this paper is to explore the feasibility and effectiveness of the MLP-KFold model in correcting rock mechanics measurements based on the available dataset. The model has demonstrated excellent data utilization efficiency and evaluation stability, which is particularly significant given the limited data scale. We agree that expanding the dataset to include a wider variety of rock types would further enhance the model's generalization ability and robustness. In our subsequent research, we plan

to conduct hydraulic fracturing experiments on rock cores of different lithologies to collect more diverse data and address the current limitations of dataset scale and diversity.

We appreciate your understanding of the challenges associated with experimental data collection in this field and hope that our current work can serve as a preliminary exploration, with more comprehensive improvements to follow in future studies.

3. Model Generalization Capability.

Explanation and modification: Thank you for raising the important issue of the model's generalization capability. We recognize the need to further validate and discuss the model's applicability across diverse geological environments and construction conditions. While the MLP-KFold model demonstrated remarkable performance on the test set with a coefficient of determination ($R^2=0.9937$), it is acknowledged that the diversity of rock types, formation conditions, and construction parameters in real-world applications could pose challenges to the model's generalization capability, and the following aspects discussed in section 5.3 in red.

(1) Cross-Validation with diverse datasets: by merging cross-validation with datasets spanning various geological settings and construction conditions with field trials in multiple locations, we can comprehensively assess the model's performance in real-world applications. This integrated method not only identifies potential weaknesses or biases in the model but also provides empirical data from different geological environments, thereby enabling targeted adjustments to improve generalization.

(2) Incorporation of additional features: to further bolster the model's adaptability, we advocate for the incorporation of additional features that encapsulate the variability in geological and construction parameters. These features may encompass rock anisotropy, formation fluid properties, and dynamic construction variables, among others. In parallel, establishing a framework for continuous model updating based on new data and feedback from field applications ensures the model evolves with emerging geological and construction challenges, maintaining its accuracy and relevance.

In summary, these strategies aim to enhance the model's generalization capability and reliability across a broad spectrum of practical applications. Future efforts will concentrate on expanding the dataset, conducting extensive field trials, and refining the model to address the intricacies of real-world hydraulic fracturing operations.

4. The content of the Conclusions is too extensive and should be more refined.
Modification: We have rewritten the conclusion based on the revision suggestions.

Format issues:
1. Table 5 is too large and should not appear in the main text. It should be placed in the appendix.
Modification: We have included Table 5 as an appendix at the end of the manuscript.

2. The font in Figure 3 is too small, but the font in Figure 4 is too large, which should be consistent with the font in the main text. The rest of the pictures also have this issue, please adjust them.

Modification: We adjusted the dimensions of Figures 3, 4, 5 and 9 to be consistent with the font in the main text.

---

## Author Comment (AC4)

Dear reviewer 2#:

We are very grateful for your comments, we have made modifications and proofreading based on the reviewer 1# and 2#, the revised parts have been marked in red in revised version, and the detail modifications are shown in detailed revision. We look forward to hearing from you.

Best regards,

Yimin Liu and Jinwu Luo.

Detailed revision:

1. The abstract and conclusions should be more concise, as it seems that there are too many words in the initial draft.

Modification: We have simplified the abstract in revised version.

2. More representative literature should be selected in the literature review.

Modification: We have removed some literature that is not strongly related to the topic of this paper, such as Guo et al. (2023).

3. The content of Chapter 3.1 is not very relevant to the main content of this paper. It is recommended to merge and rewrite Chapter 3.1 and 3.2.

Modification: We have redesigned the structure of Chapter 3, removing some content from Section 3.1 and merging the rest into Section 3.2 to make it more closely aligned with the theme of the paper.

4. This article only establishes a compensation model for in-situ stress calculation of granite core. What if the generalization ability of this model is improved?

Modification: Thank you for raising the important issue of the model's generalization capability. We recognize the need to further validate and discuss the model's applicability across diverse geological environments and construction conditions. While the MLP-KFold model demonstrated remarkable performance on the test set with a coefficient of determination ($R^2$=0.9937), it is acknowledged that the diversity of rock types, formation conditions, and construction parameters in real-world applications could pose challenges to the model's generalization capability, and the following aspects discussed in section 5.3 in red.

(1) Cross-Validation with diverse datasets: by merging cross-validation with datasets spanning various geological settings and construction conditions with field trials in multiple locations, we can comprehensively assess the model's performance in real-world applications. This integrated method not only identifies potential weaknesses or biases in the model but also provides empirical data from different geological environments, thereby enabling targeted adjustments to improve generalization.

(2) Incorporation of additional features: to further bolster the model's adaptability, we advocate for the incorporation of additional features that encapsulate the variability in geological and construction parameters. These features may encompass rock anisotropy, formation fluid properties, and dynamic construction variables, among

others. In parallel, establishing a framework for continuous model updating based on new data and feedback from field applications ensures the model evolves with emerging geological and construction challenges, maintaining its accuracy and relevance.

In summary, these strategies aim to enhance the model's generalization capability and reliability across a broad spectrum of practical applications. Future efforts will concentrate on expanding the dataset, conducting extensive field trials, and refining the model to address the intricacies of real-world hydraulic fracturing operations.

5. The MSZK and ZPZK boreholes in engineering application should not reflect the specific location, just use a certain drilling hole.

Modification: We have removed the specific locations of MSZK and ZPZK boreholes in Chapter 6.

6. Please adjust the font size of the pictures in the paper to maintain consistency.

Modification: We adjusted the dimensions of Figures 3, 4, 5 and 9 to be consistent with the font in the main text.